# Factors Affecting Canine Obesity Seem to Be Independent of the Economic Status of the Country—A Survey on Hungarian Companion Dogs

**DOI:** 10.3390/ani10081267

**Published:** 2020-07-24

**Authors:** Torda Orsolya Julianna, Vékony Kata, Junó Vanda Katalin, Pongrácz Péter

**Affiliations:** Department of Ethology, ELTE Eötvös Loránd University, 1117 Budapest, Hungary; torda.orsolya@gmail.com (T.O.J.); kata.vekony.kami@gmail.com (V.K.); juno.vanda@gmail.com (J.V.K.)

**Keywords:** dog, obesity, owner, age, activity, feeding

## Abstract

**Simple Summary:**

Obesity is an ever-growing epidemic among people and their pets. Recently, more and more studies investigated the risk factors of dog obesity, but most of them only included data from high-income Western countries. We surveyed Hungarian dog owners about their dogs’ body conditions and social and demographic factors that might affect it. Factors affecting canine body condition seem to be similar to those of high-GDP (Gross Domestic Product) countries, suggesting that these findings are more or less universal. Older dogs are more likely to be overweight/obese than younger ones. Joint activity and sports performed with the owner, even on a hobby level decrease the likelihood of obesity, and the main source of calories (kibble, human leftovers, or raw food) also had an effect on body condition. These findings support that the main risk factors of dog obesity are the ones controlled by the caretakers of dogs, therefore it is important to educate dog owners about how they can prevent the development of this condition.

**Abstract:**

For a companion dog, the most important environmental factor is the owner, who is responsible for providing everything that is necessary for the health and welfare of the dog. Still, one of the most often compromised measures of an average companion dog is its body condition, with overweight and obese animals representing a growing welfare problem around the world. Using an extensive online survey that was distributed among Hungarian dog owners, we wanted to know whether those obesogenic factors that were identified mostly in high-GDP countries’ pet dog populations would hold their relevance in a much lower average income country as well. We found several associations between the body condition of canine companions and various demographics, environmental and behavioral factors. Older dogs reportedly had an accelerating propensity for being overweight. Joint activity and performing dog sports both reduced the likelihood of being an overweight dog. Main food types also had significant associations with the body condition of dogs—meanwhile, the feeding of commercial dog food (kibble) and/or leftovers of human meals coincided with being overweight, dogs that were fed (fully, or at least partly) with raw food were less likely overweight. In the case of owner-reported behavioral problems, the food-related issues (stealing food, overeating, etc.) were clustered to a dimension together with problem behaviors such as excessive barking and overt aggression. Beyond showing a good agreement with earlier surveys on associated factors with canine obesity, our study revealed some interesting new details that could be used in the prevention of overweight problems in dogs.

## 1. Introduction

According to some estimations, the recent dog population on Earth may reach a staggering one billion animals [1]. Apart from a relatively negligible number of “true” feral dogs (such as the Dingo, [2]), the everyday existence of most dogs almost fully depends on the provision/exploitation of human resources. Although approximately 80% of the world’s dog population belongs to the “ownerless” category (the so-called pariah/village/street dogs [1]), still, hundreds of millions of dogs are being kept, largely in the well-industrialized countries, by identifiable individuals (“owners”), mainly with the function of companionship. In the case of companion dogs, it is the responsibility of the humans to fulfill all the biological needs of their animals, ranging among others from basic nutrition to the properly enriched social environment and activities and necessary medical assistance. When humans fail in one or the other area of responsible ownership, the consequences can affect negatively the welfare of dogs (e.g., [3]), the humans (e.g., [4]), or the broader environment (e.g., [5]). Being obese or overweight, which is considered as one of the most pervasive chronic conditions, leading to lessened life quality and decreased life expectancy in both humans and dogs [6], is one of those serious canine welfare problems, where the causation almost exclusively can be attributed to misguided husbandry practices by the owner. 

The relationship between companion dogs and their owners (or, in a broader sense, the anthropogenic environment) is arguably the most intimate and complex social construct among the inter-specific bonds of our species. There is a plethora of empirical evidence about the socio-cognitive capacity of dogs that is considered to be the product of human-analogous evolutionary processes since their domestication [7]. Dogs are more and more regarded as family members in Western societies [8], with an equally prestigious segment of financial expenditure both on the level of the individual owner and the societies [9,10]. Dogs recently became a popular “model organism” in various fields of empirical science (e.g., molecular genetics [11], personality research [12] and behavioral-cognitive impairments [13]), because their socio-cognitive analogies and shared everyday environmental effects with humans may provide an ecologically and evolutionarily more valid referential framework than the traditional invertebrate, rodent or primate systems. In the case of canine obesity, humans are not only the primary elicitors of this condition, but at the same time, the two species also share most of the causative factors [14], and mental/physical epidemiology [15] related to overweight/obesity problems. Therefore, canine and human obesity can be regarded as an interconnected health and welfare problem of modern societies, where empirical results about one of them can be illuminating for the other and vice versa. 

Obesity develops when energy intake exceeds energy expenditure over an extended period. It is a multi-causal condition [16], where, in the case of dogs, the genetic background of breed-specific over-eating was recently also highlighted (e.g., [17]). Another intrinsic factor, which seems to be relevant in the motivation behind particular dogs’ intense food intake is the inclination towards reward-maximizing. Depending on the research, such tendencies were described in dog breeds selected for hunting [18], or for working tasks that do not require constant feedback from the handler (“independent working dogs” [15]). Regarding other causative interactions, the role of sterilization (spaying and neutering) has a firmly established role [19] and there are indications that dogs’ age and body condition are not independent of each other either. From the aspect of man-made canine obesity, feeding-related, and physical activity-related factors could be labeled as decisive [20]. Different authors highlight that neglectful or erroneous husbandry strategies chosen by the owners could be the primary causes of canine obesity [21,22], since most of the obesity risks in dogs are related to the chosen way of life, habits, and values of the people [23], among which the type and amount of nutrition and physical activities may play an especially important role.

Some intrinsic and environmental factors that are associated with the body condition of dogs are highly suitable for empirical testing (e.g., genetic predisposition [24], intake of energy/nutritional needs [25], associations with stressful environment [26]), and the reliable experimental approach of these often involves the employment of laboratory dog populations (e.g., [27]). However, the life history of companion dogs is strongly affected by the anthropogenic environment that is provided by the owner (e.g., [28]), and this creates the need for a different research approach when one is interested in extrinsic factors of canine obesity. Large scale surveys based on either the clientele of veterinary examinations (e.g., [19,29]), or questionnaires that were completed by the owners (e.g., [22]) are at the forefront of studying the presence and causation of overweight problems in the companion dog population. So far, several factors were highlighted in association with the body conditions of companion dogs: keeping environment [30], daily activity/exercise [31], type and timing of feeding [19], and provisioning of treats [22]. These results are typically based on participants (owners and their dogs) living in industrialized societies with high GDP/capita ratios (e.g., Australia [29], France [32], and the USA [19]). Among others, the average income level of families is considered to be an important factor of the so-called “obesogenic environment” [33,34], thus one would expect that the socio-economic conditions in a country would affect the body condition of companion animals as well. A multi-national survey [23] found that dogs that were owned by citizens of lower-income European countries were more likely to be overweight/obese than the dogs living in very high-income European countries. According to the hypothesis of the authors, this can be caused by the higher awareness among dog owners about the negative consequences of obesity in the higher-income countries. 

In this study, we performed a questionnaire survey among Hungarian dog owners. Besides asking them to report about the body condition of their dogs, we focused on particular social factors (keeping conditions, joint activity, feeding, providing treats, and behavioral problems) that may affect the body condition of the dogs and at the same time under significant human control. According to the IMF’s statistics, based on the GDP per capita data Hungary’s economic status is in the lower half of the countries (31st from 49, [35]). Meanwhile, in Hungary, the popularity of dog ownership (expressed as the ratio of households with at least one dog) is high [36], and human obesity reaches an alarming ratio among Hungarians [37], we are not aware of any scientific study concerning overweight canines and its associated factors from this country. From the aspect of the relatively low GDP per capita of Hungary, this study can be considered as the first detailed survey of the factors affecting canine body condition in a non-high income European country, providing a useful addition and validation at the same time for the already established associations between canine overweight problems and some of the human-controlled factors behind them. 

We conducted an internet questionnaire study among Hungarian dog owners to see how certain external factors affect obesity in family dogs. As the average income of Hungarians [38] is considerably lower than in many countries where earlier dog obesity studies were conducted in, and there can be also differences among Hungarians and dog owners of Western societies in their attitudes toward their pets [12,39], we wanted to find if the earlier found effectors behind canine obesity are still effective among Hungarian dogs as well.

If we find the expected correlations, this would mean that the well-studied and proven methods of preventing and treating obesity could work just as well in the case of Hungarian family dogs.

We also studied in a more detailed manner the different feeding methods, choices of food, and the frequency of feeding the dog with these foods. We also wanted to know how different behavioral problems co-occur with canine obesity and to compare them to behavioral traits associated with human obesity.

Our main goal is to expand our knowledge about obesity and the underlying causes and to extend the validity of previous findings to countries with low GDP for having a more accurate knowledge to prevent and treat this ever increasing health risk factor in both people and their companion animals.

## 2. Materials and Methods

### 2.1. The Survey

We surveyed the owners’ habits in regard to the keeping and feeding of their dogs as factors that can be associated with their body condition. The questionnaire was created in the Hungarian language and it was distributed via e-mail and on social media, mainly through dog-related Facebook groups and pages. No form of incentive was offered for participation in our survey, and the questionnaire was anonymous. 

The questionnaire consisted of five parts. In the first part, we asked basic data about the dog (e.g., age, sex, and reproductive status) which are known to have an effect on body condition [19,40,41], as well as the possible health issues.

Items in the second part reflected the dog–owner relationship, such as the role of the dog in the owner’s life or the time spent together.

As it is known that certain behavioral traits can be associated with condition and obesity both in humans and dogs [15,26,42], in the third part we asked about behavioral problems.

The fourth part of the questionnaire contained a series of questions about the feeding of the dogs, investigating both the main types of food and their frequency in the dogs’ diet, plus the type and use of rewards in training.

Physical activity has a great effect on the body condition of the dog [6,22], so the fifth part of the questionnaire consisted of questions regarding the type and frequency of joint activities of the dog-owner dyads.

We used three independent items from the questionnaire for the assessment of the dogs’ body condition. (i) We asked the owners to use the three-level scoring system (underweight/thin, correct weight, overweight/obese) to assess their dog’s condition by palpation (see also [15]); (ii) we asked their holistic opinion about the dog’s body condition (three-point scale); and (iii) we asked if their veterinarian ever mentioned that their dog was over- or underweight.

### 2.2. Subjects

Participation in the survey was voluntary and anonymous, 1448 responses were recorded in total, and each response was about one dog (in the case of multi-dog households, the owner could fill separate questionnaires for each dog). The dogs were of various breeds, ages (mean = 4.2, SD = 3.09), sex and reproductive statuses (males: *n* = 662, females: *n* = 786, from which neutered males: *n* = 305, neutered females: *n* = 517, respectively).

### 2.3. Variables

Table 1 and Table 2 show the detailed scoring system for the variables that we used in the statistical analysis. From each of the five parts of the questionnaire, we derived variables to be used as independent variables. As dependent variables, we used three measures of the dog’s body condition: (i) the body condition score according to the three-level palpation scoring system; (ii) the owner’s holistic opinion about their dog’s body condition; (iii) the owner’s anamnesis about their veterinarian’s opinion if the dog was underweight, normal or overweight.

### 2.4. Data Analysis

We used IBM SPSS (version 22.0, Armonk, NY, USA) and R statistical software (R Development Core Team, 2015) in RStudio (RStudio Team, 2018) with packages ca, FactoMineR, dplyr, MASS, and emmeans for the statistical analyses.

At first, we used Pearson’s correlation to compare the three methods of assessing the dogs’ body condition. We used multiple correspondence analysis (MCA) to reduce the number of behavioral issue-related variables and found the top dimensions that explain most of the variability. The association between the independent factors and dogs’ body condition was analyzed with ordinal regression.

## 3. Results

According to the three-point palpation method, 4.7% of the dogs were overweight (*n* = 68), 86.05% had ideal (normal) body condition (*n* = 1246) and 9.25% were underweight (*n* = 134). Owners of 12% of the dogs (*n* = 174) were told at least once by their vet that their dog is overweight, 78.9% were never told that there’s a problem with their dog’s body condition (*n* = 1142) and 9.1% were told that their dog is underweight (132). Nine percent of owners thought that their dog is overweight (*n* = 130). We found that the three methods for assessing the dogs’ body condition strongly correlate with each other (Table 3). From this point on, each of the following analyses were performed on the body condition scores derived by the palpation method, as this can be considered to be the most empirical approach from the three assessment types.

Age had a significant association with body condition: a post-hoc test showed that dogs above 10 years were more likely to be overweight than all age groups under 5 years (age group 3–age group 5: est. = −0.79, Z = −2.909, *p* = 0.0298), and dogs above 5 years were also more likely to be overweight than all age groups under 2 years (age group 2–age group 4: est. = −0.94, Z = −4.157, *p* = 0.0003)(Figure 1).

Activity with the dog turned out to be another important factor: both the time spent actively together with the dog and any sporting activity had a significant association with body condition. Dogs that spent less than one hour actively together with the owner (walking, running, playing) a day were more likely to be overweight than dogs that spent more than one hour (est. = −0.804, Z = −3.146, *p* = 0.009). Only two owners responded “none” to this question. There was no significant difference between groups above one hour of joint activity (Figure 2A). Dogs who are engaged in dog sports even on a hobby level were less likely to be overweight than dogs who do not do sports (est. 1.042, Z = 4.996, *p* < 0.0001). There was no significant difference between the body conditions of dogs that perform sports at a hobby or competitive level (Figure 2B).

We also found that feeding had a significant association with body condition: dogs that were never fed with commercial dog food were less likely to be overweight than dogs that were fed commercial dog food with any frequency (est. = −0.501, Z = −2.635, *p* = 0.0084) (Figure 3A) and dogs that were never given human food (mainly leftovers) were less likely to be overweight than dogs that were fed human food with any frequency (est. = −0.636, Z = −3.967, *p* = 0.0001) (Figure 3B).

Feeding uncooked or raw food had an opposite effect than the other food types: dogs that never ate raw food were more likely to be overweight than dogs that were fed raw food with any frequency (est. = 0.399, Z = 2.552, *p* = 0.0107) (Figure 4A). We also found that dogs that were fed human food but also ate raw with any frequency were less likely to be overweight than dogs that were fed human food but not raw (est. = 0.707, Z = 3.542, *p* = 0.0004), and there was no significant difference between the body condition of dogs eating both human food and raw and dogs that never ate human food (Figure 4B). We did not find this effect of feeding raw in dogs that were also fed with commercial or non-commercial dogfood (the number of dogs fed with combinations of raw and some other food type is shown in Table 4).

We found that the top two dimensions of MCA explained 74.5% of all variance in the case of the behavior problem-related answers. The first dimension (Dim1) is characterized by hyperactivity, jumping on people, fear, being noise sensitive and/or disobedient, while the second dimension (Dim2) is characterized by excessive barking, aggression, fear, and eating-related problems (Figure 5). We used scores obtained for these dimensions in the analyses.

When investigating the associations between body condition and behavior problems, we found the following significant interactions: overweight dogs of owners who saw them as animals or tools of work scored significantly lower on the first dimension of behavior problems (est. = 11.936, Z = 3.595, *p* = 0.0009); dogs with normal body condition who received non-food reward during training (e.g., praise) scored significantly higher on the second dimension of behavior problems (est. = −6.65, Z = −2.374, *p* = 0.0176); underweight dogs that never received food reward showed a non-significant trend towards scoring higher on the second dimension of behavior problems (*p* = 0.0679).

## 4. Discussion

In an extensive, online survey that was distributed among Hungarian dog owners, we found several associations between the body condition of canine companions and various demographic, environmental, and behavioral factors. Older dogs reportedly had an accelerating propensity for being overweight, which started above five years of age and became the strongest over ten years of age. Joint activity and performing dog sports both reduced the likelihood of being an overweight dog and remarkably, in the case of these factors we found no difference between the associations of body condition with moderate or high levels of activity. Main food types also had significant associations with the body condition of dogs—meanwhile, the feeding of commercial dog food (kibble) and/or leftovers of human meals coincided with being overweight, dogs that were fed (fully, or at least partly) with raw food were less likely overweight. In the case of owner-reported behavioral problems, the food-related issues (stealing food, overeating, etc.) were clustered to a dimension together with problem behaviors such as excessive barking and overt aggression. Body condition of dogs had complex associations with various factors including the behavioral dimensions, with a general tendency where overweight dogs showed slightly fewer behavioral issues (in the owners’ opinion), meanwhile, the normal and underweight dogs scored somewhat higher on those behavioral dimensions, which also included the feeding-related problems. 

Among the main goals of this study was to investigate whether the associations that were found between the body condition of dogs and various extrinsic factors in other investigations, mostly performed in countries with high or very high GDP, would be also true in a low-medium GDP European country, Hungary. This aspect perhaps [23] offers the most comprehensive wealth of data for comparison with our findings. Based on a convenience sample of more than three thousand questionnaire responses from ten European countries, the ratio of overweight/obese dogs was the lowest in the countries with the highest GDP, meanwhile, owners reported a significantly higher occurrence of overweight/obese dogs from the medium and low-income countries [23]. It is an interesting aspect of our present Hungarian survey that we recorded a much lower occurrence of overweight/obese subjects than [23]. Our sample is also large (almost 1500 entries) and similarly to [23] we also used a convenience sample of internet-responders. We should keep it in mind also that our sample contains a much higher number of participants from a single country than we found in [23] (Hungary was not included in that survey). One could assume that the large discrepancy between the reported ratios of overweight/obese subjects could be due to the different scoring systems used in the two studies. We asked the owners to use a three-grade palpation assessment, [23] used the five-grade method. It is possible that meanwhile, the three-grade method underestimates the body condition of dogs, the five-grade system may inflate somewhat the ratio of being an overweight/obese dog. This assumption can be partly supported with the positive correlation between the three scoring systems we tested in our survey, where at the same time the palpation method provided the lowest ratio of overweight/obese subjects, compared to the owners’ holistic opinion and the anamnesis of previous comments made by the veterinarians. Therefore, it is safe to say that the 4% of heavier-than-ideal dogs found in this study (by using the three-grade palpation method) would be closer to “obese” in the five-grade palpation system than to slightly overweight. This prevalence is very similar to the proportion of obese adult dogs reported by [19] in their large US sample based on private veterinary practices’ data.

Our results show a close alignment with other international surveys regarding the associations between canine body condition and environmental factors such as age and activity. Similar to our results, it was found in other European countries [23,32] and in the United States [19] that obesity in dogs is more frequent among the older canine companions. Besides the confirmation of this result in our Hungarian sample, we can highlight as a further detail that obesity shows an accelerating prevalence as the dogs pass five years of age, and the age cohort between 5 and 10 years is already significantly more obese than the younger dogs. In the future, further research could illuminate the possible interplay between the breed-dependent onset of old age in dogs [43] and the age-dependent elevation of obesity risk in dogs. 

Although the joint physical activity related to dog ownership received high interest from the aspect of human health (e.g., [44]), in our survey we concentrated on the associations between canine obesity and the various activities that require the contribution of the dog owner. This latter aspect, although instinctively seems important, was rather rarely included in studies before (but see [23] and for example [45]). We found that the least active dogs were most likely to be the obese ones, and importantly, even a moderate amount of joint activity/exercise with the owner significantly lessened the risk of obesity—and from this aspect, there was no difference between doing dog sports or performing less organized activities (such as walks). Meanwhile one could assume that there is a considerable overlap between the subjects that do “joint activity with the owner” and perform “sports”, these results warrant that canine obesity probably could be avoided successfully with a slightly more than minimal effort from the owner’s side with including regular and not even strenuous sessions of being active with his/her canine companion. 

Perhaps not surprisingly, in our study, the type of food what owners provided to their dogs was also in strong association with dogs’ body condition. Feeding as a causative factor is in the forefront of the scientific literature about canine obesity (e.g., [46]). Although there can be differences between the emphasis of the different studies, sharing table scraps with the dogs [19,23], feeding them with semi-moist commercial foods [19], and feeding them with high-energy/low raw fiber content foods [46] are commonly mentioned as associated factors of obesity. As an interesting contrast, [23] mentions also that in European countries dogs were more often found as underweight when they were fed with “homemade” food—with no further information whether in that study this meant raw or cooked food for the dogs. According to our results, dogs in Hungary are more likely obese when they are fed on either commercial dog food or table scraps—however, dogs that were fed with raw food items were less likely obese. Interestingly, even lower frequencies of feeding with raw food has a counterbalancing effect against obesity in the case of parallel provisioning of table scraps. This association was not found in the case of dogs that were fed with commercial dog food and raw food. 

Provisioning companion dogs (and cats) with raw food instead of commercial heat-treated food (“kibble”) has a growing momentum in the last decades [47]. Among the reasons for feeding raw, we can mention the growing mistrust in commercial kibble makers, or the assumed/proven positive health consequences on pets [48]. Concerns about feeding unprocessed raw food items (and especially commercial raw dog food) are accumulating from the aspect of the potentially high prevalence of pathogenic microbes [48], and because of the likelihood of imbalanced nutrition through raw feeding [49]. Curiously, currently, there is almost no available publication about the possible association between dogs’ body condition and feeding raw—for example, [22,50] did not find any association between being overweight and being fed on raw, [46] only advises raw feeding when its appropriateness and demand was approved by a professional, and [23] did not specify whether the “homemade dog food” in their study was raw or not. In our study, we did not survey whether Hungarian dog owners were feeding to their dogs human-grade raw food or commercially available raw dog food. In our study, both the feeding of table scraps and processed commercial dog food (kibble) turned out to be in positive association with canine obesity. Meanwhile, leftovers of human food are often mentioned as being obesogenic [19,23], feeding the dogs with commercial kibble itself is surprisingly lacking from the published causes of obesity. An indirect reason for this can be the predominance of commercial dog food feeding in the industrialized societies (i.e., the lack of suitable non-kibble fed control group). An exhaustive study [51] on Australian pet dogs found no difference among the body condition of dogs fed with various types of commercial foods. We assume that in the case of our study, kibble-feeding might enhance the probability of obesity among Hungarian dogs because the owners can misjudge the required portions easier with this food type, or because in a country with lower GDP there is less chance for the owners to purchase good quality commercial dog food. We also should mention that because of the bewilderingly wide assortment of commercial dog food types fed to the subjects according to our survey, it was not feasible to perform further analysis on the association between body condition of dogs and particular (price range) categories of kibble. Another limitation regarding the results about the association between canine body condition and feeding with leftover human food can be that depending on the particular country, the diet of the humans (thus the caloric content of the leftovers) can also strongly vary. Further comparative studies would be necessary to analyze this possible factor behind the consequences of feeding table scraps to companion dogs.

Meanwhile, in the case of humans, the behavioral [52] and psychological [53] problems are often studied factors as being associated with obesity—in dogs, the possible parallels between body condition and behavioral problems or personality traits are rarely discussed. It was found for example that overweight dogs show a negative cognitive bias, which can be interpreted as a “win-stay” search strategy, with a possible reward-maximizing effect [15]. In the present survey, we requested a more detailed anamnesis of the various behavioral problems of their dogs from the dog owners. Interestingly, we did not find an association between feeding-related problems (e.g., stealing food, uncontrolled overeating, etc.) and being obese. Contrary, the behavioral dimension that included feeding-related problem behaviors showed higher values in the case of normal and underweight dogs, especially in those cases when owners rather opted for non-food rewarding. A possible explanation for this can be that owners react with food restriction when they notice that their dogs show an unwanted level of food-motivation—however, this association may also have the opposite causative relationship.

## 5. Conclusions

In the case of dogs, most obesogenic factors are under human control or influence. Our survey showed that companion dogs in a middle-low income European country, Hungary, in general, are under the same environmental effects regarding their body condition as the dogs in higher-income countries. These results have a positive consequence on the generalizability of earlier results on canine overweight problems because it is important that dog owners and professionals can rely on studies, which otherwise do not involve international samples. The relevance of dogs’ age, level of activity, type of feeding when it comes about body condition are such basic knowledge that is freely and easily understandable for the wider public—therefore, dog owners can formulate their own decisions when trying to avoid the onset of being overweight in their canine companions. As our study highlighted also some new and previously not shown or rarely investigated factors that may enhance or decrease obesity in dogs (e.g., the interplay between feeding raw and processed food items; the association with behavioral problems), further investigations are encouraged to be performed. Especially the connections between behavioral/cognitive problems and the dogs’ body condition seems like a promising field of research, as there are plenty of parallels in the human literature, but these associations have not been thoroughly investigated among dogs yet. 

## Figures and Tables

**Figure 1 animals-10-01267-f001:**
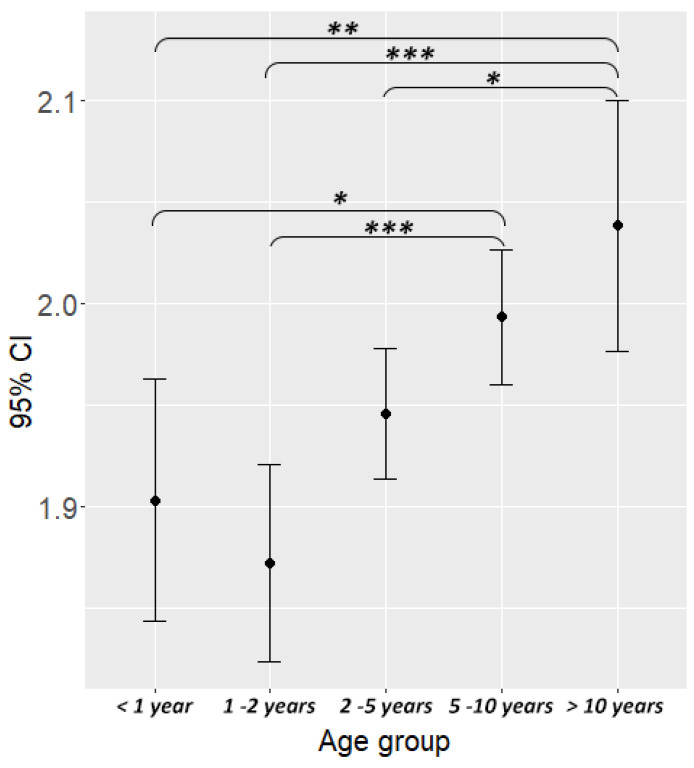
Dogs’ age had a significant association with body condition: dogs in the oldest age group (>10 years) were more likely to be obese than dogs in the first three age groups and dogs in the fourth age group (between 5–10 years) were more likely to be obese than dogs in the first two age groups. Asterisks indicate level of significance: * *p* < 0.05; ** *p* < 0.01; *** *p* < 0.001.

**Figure 2 animals-10-01267-f002:**
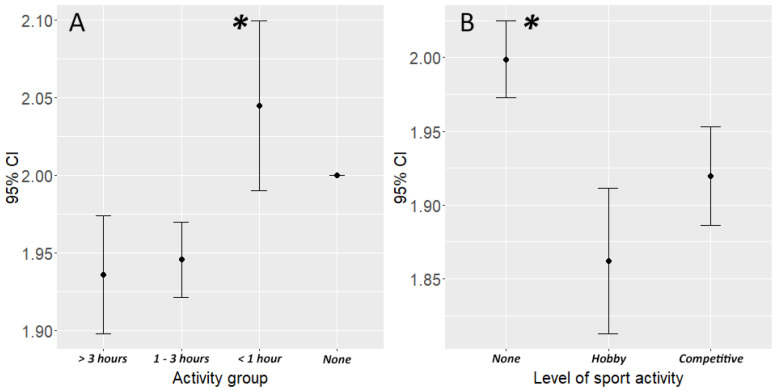
The associations between daily activity and sport activity with the body condition of the dogs. (**A**) Dogs that spent at least one hour a day actively with their owners were less likely to be overweight and (**B**) dogs that are engaged in sports were less likely to be overweight. Asterisk denotes those groups that significantly differ from the others (*p* < 0.05).

**Figure 3 animals-10-01267-f003:**
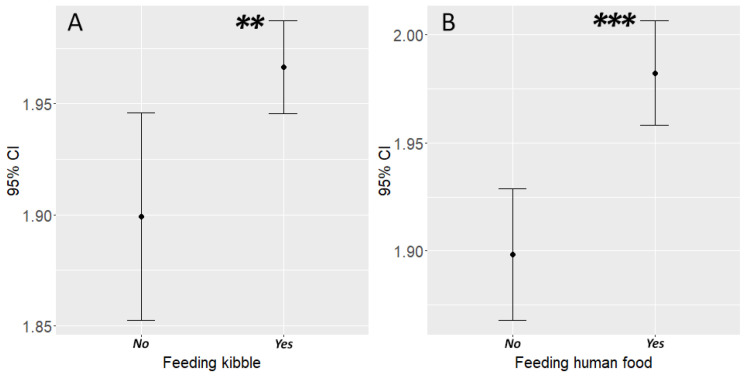
The association between feeding and body condition of dogs. (**A**) Dogs that eat commercial dogfood were more likely to be obese and (**B**) dogs that eat human food were more likely to be obese. Asterisks indicate the level of significance: ** *p* < 0.01; *** *p* < 0.001.

**Figure 4 animals-10-01267-f004:**
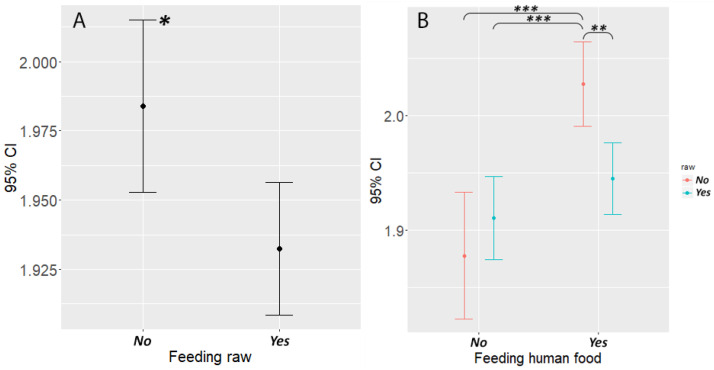
The association between feeding uncooked or raw food and the body condition of the dog. (**A**) Dogs that were fed raw food with any frequency (even less than once a week) were less likely to be overweight and (**B**) dogs that were fed uncooked or raw food were less likely to be overweight even if they were also fed with human food. Asterisks indicate the level of significance: * *p* < 0.05; ** *p* < 0.01; *** *p* < 0.001.

**Figure 5 animals-10-01267-f005:**
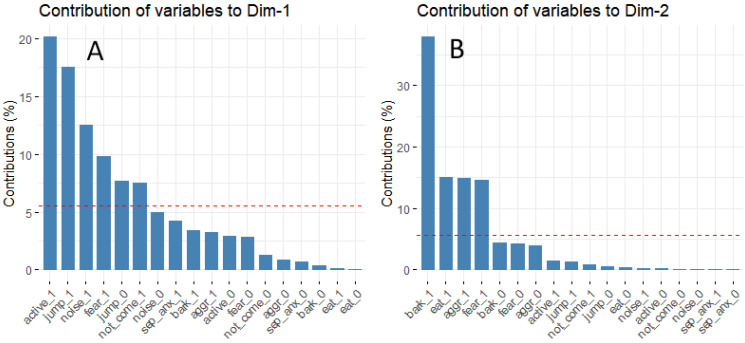
Contribution of each behavior problem to the top two dimensions of MCA. (**A**) Dim1 is characterized by hyperactivity, jumping, noise sensitivity, fear, and not coming when called and (**B**) Dim2 is characterized by excessive barking, eating-related problems, aggression, and fear. The red dashed line on the graph above indicates the expected average value if the items’ contributions would be uniform to the dimensions.

**Table 1 animals-10-01267-t001:** The independent variables derived from each part of the questionnaire. In the last column, we indicated the levels of measure the given variable could take in the statistical analysis.

Name of Variable	Description	Measure
Demographic Data		
Dog Age	Age of the dog divided into five age groups: below 1 year; 1–2 years; 2–5 years; 5–10 years; above 10 years	1–5
Place of Keeping	Where does the owner mainly keep his/her dog: inside (in an apartment or house), both outside (garden or kennel) and inside the house; in the garden; in a kennel	1–4
Dog–Owner Relationship		
Dog Role	What is the dog’s role in the owner’s life: like a child or family member; friend, playmate or colleague; working or domestic animal	1–3
Active Time	The amount of time the owner actively spends with the dog in a day (training, playing, walking, etc.): more than 3 h; 1–3 h; less than 1 h; none	1–4
Behavioral Problems		
Separation anxiety		0; 1
Aggression (towards people or other dogs)		0; 1
Food-related problems (stealing food, devouring food)		0; 1
Fearful		0; 1
Jumping on people		0; 1
Noise sensitivity		0; 1
Excessive barking		0; 1
Hyperactivity		0; 1
Not coming back when called		0; 1
Reward		
Reward Type	If the owner ever uses food as a reward during training	0; 1
Feeding		
Commercial	If the owner ever feeds commercial dog food to the dog	0; 1
Human Food	If the owner ever feeds human food to the dog	0; 1
Home-made	If the owner ever cooks specifically for the dog	0; 1
Raw Diet	If the owner ever feeds a raw diet to the dog	0; 1
Main Food Type	The main type of food the owner gives to the dog: only commercial; never commercial; both commercial and other	1–3
Activity		
Physical Activity	Level of physical activity: none; hobby; sport	1–3

**Table 2 animals-10-01267-t002:** The variables used to measure the dog’s body condition (dependent variables).

Body Condition		
**Palpation**	The dog’s condition according to the three-level scoring system. Underweight: It is easy to touch and see the contour of the ribs, the lumbar vertebrae, and the pelvic bones. The muscular mass is reduced and the body fat is almost undetectable under the skin. The dog has a markedly visible waist section. Normal: The ribs are easy to touch; they are covered by a thin layer of fat under the skin. The waist section is well visible. Overweight: It is difficult, or impossible to locate the ribs, because of the thick layer of fat under the skin. The waist section is almost undetectable.	1–3
**Owner Opinion**	The owner’s holistic opinion on the dog’s body condition: overweight; not overweight.	1–2
**Veterinarian Opinion**	The veterinarian’s opinion about the dog’s body condition: underweight; normal; overweight.	1–3

**Table 3 animals-10-01267-t003:** Pair-wise correlations between the three methods of assessing a dog’s body condition.

		Condition According to the Three-Level Scoring System	Condition According to the Owner	Condition According to the Veterinarian
Condition According to the Three-level Scoring System	Pearson correlation	1	0.364	0.296
	Sig. (2–tailed)		<0.0001	<0.0001
Condition According to the Owner	Pearson correlation		1	0.407
	*p* (2–tailed)			<0.0001
Condition According to the Veterinarian	Pearson correlation			1
	*p* (2–tailed)			

**Table 4 animals-10-01267-t004:** Number of dogs fed with different combinations of raw and other food types.

Kibble	Raw	N
No	No	23
No	Yes	235
Yes	No	596
Yes	Yes	594
Human food	Raw	N
No	No	180
No	Yes	302
Yes	No	439
Yes	Yes	527
Non-commercial	Raw	N
No	No	263
No	Yes	335
Yes	No	356
Yes	Yes	494

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
