# Peer review of "Factors Affecting Canine Obesity Seem to Be Independent of the Economic Status of the Country—A Survey on Hungarian Companion Dogs"

_animals, 2020, doi:10.3390/ani10081267_

Round 1

Reviewer 1 Report

Animals-865507

Comments and Suggestions for Authors

The authors surveyed Hungarian dog owners to get information about the social and demographic factors that can affect their dogs' body conditions. This type of data has been collected in previous papers for other European countries with high-GDP, while the authors provide data concerning a country with lower GDP.

The paper provides interesting information regarding the correlation between dog obesity and various factors such as age, the type of activity carried out by dogs and the food taken. The field of research analysed is interesting.

General comments:

The paper is well written, the writing is smooth and simple, and English is very readable. Overall, the paper is interesting.

Specific comments:

The association with the type of food (kibble, human leftovers, raw food) is a bit weak. It would be more appropriate to evaluate precisely the quality of the kibble (perhaps through the price) and it must be considered that the composition of the human leftovers can change in different countries. The authors may mention this limitation in the discussion.

Line 18: Add the point at the end of the simple summary.

Lines 42, 44, 50, 63, 86, 92: Add the closing parenthesis.

Line 143: Is asked what are the health conditions of dogs?

Line 167: Remove the point after “Table 1”.

Line 213: Replace “2a” with “2A”.

Line 216: Replace “2b” with “2B”.

Line 224: Replace “3a” with “3A”.

Line 226: Replace “3b” with “3B”.

Line 228: Replace “Figure 3” with “Figure 3”.

Line 233: Replace “4a” with “4A”.

Line 237: Replace “4b” with “4B”.

Figure 4: Add the letters A and B.

Line 246: Replace “Table 4” with “Table 4”.

Line 247: Is the sentence incomplete?

Lines 401, 428, 444: Use the italics for the name of the species.

Author Response

Thank you for the supportive evaluation of our paper. 

We performed all the suggested changes regarding the formatting of the paper. Two specific comments we also took into consideration and altered the text accordingly, these comments and our responses we copy to this field below:

Reviewer 1: The association with the type of food (kibble, human leftovers, raw food) is a bit weak. It would be more appropriate to evaluate precisely the quality of the kibble (perhaps through the price) and it must be considered that the composition of the human leftovers can change in different countries. The authors may mention this limitation in the discussion.

RESPONSE: Thank you for this comment, we added these concerns as limitations / research ideas for the future to the Discussion.

Reviewer 1: Line 247: Is the sentence incomplete?

RESPONSE: We deleted this line – it was a former sub-chapter title, but we decided not to use these in the final version of the text. It was left there accidentally.

Reviewer 2 Report

The paper is well structured and written, the conclusions are supported by the analysis of the data presented.

The animals used in the study represent a group large enough to give significance to the analysis.

The results are clearly presented and convincing.

The full text is included below:

In this study, the authors observed that environmental factors have an impact on morphological and physiological conditions of a group of a Hungarian companion dog.

The existing epigenetics studies supporting the claim that the environment represents a factor capable of modifying the phenotype. At the same time, behavioural studies have long-held that dog and owner constitute a whole and that the lifestyle of both significantly influences the risk of obesity.

However, it is interesting how the authors consider the owner to be an environmental factor.

Although the authors have dealt with topics already present in literature such as the dogs' overweight/obese related to GDP, the article enriches state of the art and the conclusions are unusual.

General comments:

The paper is well structured and written; the analysis of the data presented supports the conclusions.

The animals used in the study represent a group large enough to give significance to the analysis.

The data are presented clearly, and they are convincing.

Specific comments:

The manuscript is simple to understand, the claims are convincing, and the linearity of the non-speculative conclusions are appreciable.

Author Response

Thank you for the positive evaluation of our manuscript.